# Molecular Imaging Diagnosis of Renal Cancer Using ^99m^Tc-Sestamibi SPECT/CT and Girentuximab PET-CT-Current Evidence and Future Development of Novel Techniques

**DOI:** 10.3390/diagnostics13040593

**Published:** 2023-02-06

**Authors:** Octavian Sabin Tataru, Michele Marchioni, Felice Crocetto, Biagio Barone, Giuseppe Lucarelli, Francesco Del Giudice, Gian Maria Busetto, Alessandro Veccia, Arturo Lo Giudice, Giorgio Ivan Russo, Stefano Luzzago, Mattia Luca Piccinelli, Mihai Dorin Vartolomei, Gennaro Musi, Matteo Ferro

**Affiliations:** 1Institution Organizing University Doctoral Studies (I.O.S.U.D.), George Emil Palade University of Medicine, Pharmacy, Science, and Technology of Targu Mures, 540142 Targu Mures, Romania; 2Department of Medical, Oral and Biotechnological Sciences, G. d’Annunzio, University of Chieti, 66100 Chieti, Italy; 3Urology Unit, “SS. Annunziata” Hospital, 66100 Chieti, Italy; 4Department of Urology, ASL Abruzzo 2, 66100 Chieti, Italy; 5Department of Neurosciences and Reproductive Sciences and Odontostomatology, University of Naples “Federico II”, 80131 Naples, Italy; 6Department of Precision and Regenerative Medicine and Ionian Area, Urology, Andrology and Kidney Transplantation Unit, University of Bari, 70124 Bari, Italy; 7Department of Maternal Infant and Urologic Sciences, “Sapienza” University of Rome, Policlinico Umberto I Hospital, 00161 Rome, Italy; 8Department of Urology and Organ Transplantation, University of Foggia, 71122 Foggia, Italy; 9Urology Unit, University of Verona, Azienda Ospedaliera Universitaria Integrata Verona, 37126 Verona, Italy; 10Department of Urology, University of Catania, 95124 Catania, Italy; 11Department of Urology, European Institute of Oncology (IEO) IRCCS, 20141 Milan, Italy; 12Department of Oncology and Hemato-Oncology, University of Milan, 20141 Milan, Italy; 13Department of Urology, Medical University of Vienna, 1090 Vienna, Austria

**Keywords:** ^99m^Tc-sestamibi SPECT, girentuximab PET-CT, molecular imaging, renal cancer

## Abstract

Novel molecular imaging opportunities to preoperatively diagnose renal cell carcinoma is under development and will add more value in limiting the postoperative renal function loss and morbidity. We aimed to comprehensively review the research on single photon emission computed tomography/computed tomography (SPECT/CT) and positron emission tomography computed tomography (PET-CT) molecular imaging and to enhance the urologists’ and radiologists’ knowledge of the current research pattern. We identified an increase in prospective and also retrospective studies that researched to distinguish between benign and malignant lesions and between different clear cell renal cell carcinoma subtypes, with small numbers of patients studied, nonetheless with excellent results on specificity, sensitivity and accuracy, especially for ^99m^Tc-sestamibi SPECT/CT that delivers quick results compared to a long acquisition time for girentuximab PET-CT, which instead gives better image quality. Nuclear medicine has helped clinicians in evaluating primary and secondary lesions, and has lately returned with new and exciting insights with novel radiotracers to reinforce its diagnostic potential in renal carcinoma. To further limit the renal function loss and post-surgery morbidity, future research is mandatory to validate the results and to clinically implement the diagnostic techniques in the context of precision medicine.

## 1. Introduction

In the United States, it is estimated that, in 2023, there will be approximately 81,800 new cases and 14,890 new cancer-related deaths related to kidney and renal pelvis cancers [1]. The recent widespread of imaging techniques designed to diagnose renal masses [2] lead to an incidental and small tumors identification with highly diverse imaging features with an increased variability of their histology [3,4]. The incidental small mass tumors identified among renal cancers are renal cell carcinoma (RCC) and are around 70–80% [5,6]. Subtypes of RCC have been identified to have different aggressiveness and metastatic potential [7,8]. Almost three quarters of the RCC encountered subtype is clear cell (cc) RCC (approx. 75%), then papillary (pap) RCC (approx. 15%), and the chromophobe (ch) RCC (approx. 5%) [6]. There are two subtypes that have potential for metastatic spread (ccRCC and type II papRCC). Type I papRCC and chRCC and hybrid oncocytic chromophobe tumors (HOCTs) are considered less aggressive [7]. Benign tumors (oncocytomas and angiomyolipomas (AMLs)) are also often incidentally identified in preoperative imaging (up to 20% of clinically localized renal masses) [9,10]. Lately, there has been a reduction in the postsurgical identification of benign lesions [11], and some of the predictors for benign histology have been identified as female gender, low body mass index and low volume tumor [12]. Renal biopsy can preoperatively distinguish between benign and cancerous lesions [13]. The presence of complications and some improved diagnostic rate for small tumors can still cause the urologist’s and patient’s decision to be affected by the poor negative-predictive value of renal biopsy [14], even if in some studies the complications, seeding rate [15] after biopsy and high diagnostic rate [16] are manageable by fine needle aspiration, computed tomography (CT) [17] or magnetic resonance imaging (MRI) [18] guidance and regardless of the hospital volume [19]. It is known that a percentage of renal masses surgically removed are related to benign tumors [10] or an indolent disease [20]. Therefore, a better characterization of aggressive diseases prior to surgery using novel imaging techniques, biomarkers and antibodies can reduce the number of surgeries limiting the burden of complications, morbidity and financial costs.

The aim of this review is to comprehensively assess the current evidence of the use of molecular imaging through novel devices such as SPECT and PET-CT and ^99^mTc-sestamibi and girentuximab, to discuss the results of the trials and to enhance urologists’ and radiologists’ knowledge of the current research pattern.

## 2. Materials and Methods

In order to have a clear picture of the current evidence on how radiopharmaceutical agents and imaging can diagnose renal cancer, can identify different subtypes of renal lesions, and can explore the near possibility of entering in clinical practice, we have developed a review focusing our research on the mentioned aspects. We have used PubMed database to identify original research on these topics from the last ten years. The keywords used were ”kidney neoplasm”, ”renal tumor”, ”renal mass”, ”molecular imaging”, ”positron emission tomography”, ”single photon emission tomography”, “^99m^Tc-sestamibi SPECT” and “girentuximab PET-CT”. We have included articles up to November 2022 from the last ten years and the evidence exploring metastatic renal cancer or follow-up after treatment were excluded from the analysis. The evidence was screened by title and abstract by two independent reviewers (B.B. and F.C.) and the articles included were analyzed further following the approval of authors.

## 3. Results

After searching the PubMed database, we identified 263 research articles, of which 13 were selected according to the set criteria and were published in the last ten years. Data from one ongoing clinical trial was also evaluated. In Figure 1, we have embedded the obtained results.

### 3.1. Molecular Imaging

Molecular processes of the organism can be characterized by special molecules. These antibodies or peptides are used to carry radionuclides, making the targeted organ visible on imaging techniques such as positron emission tomography (PET) or single photon emission computed tomography/computed tomography (SPECT/CT). The combination of SPECT and PET with CT have raised the diagnostic accuracy of both PET and SPECT. There are differences among PET/CT and SPECT/CT that the clinicians will have to be aware of. PET/CT seems to bring more quantitative information than SPECT/CT and appears to have better results. PET/CT can provide better image resolution, less attenuation and disperse artifacts, higher sensitivity and better and multiple radiotracers. However, PET also comes with a higher cost than SPECT and the radionuclides used are cheaper, have a longer half-life, and this can lead to better targeting potential, resulting in a higher ability to describe the biological processes [21,22,23]. Through PET, the patient experiences radiation levels comparable to that of a CT scan [24]. The selection of the tracer is critical in terms of availability, physical and radiopharmaceuticals characteristics [24,25]. The stability (at certain pH, light, temperature) and the bio-distribution of the radionuclide are important factors that will influence its in vivo decomposition [26]. SPECT radionuclides are gamma-emitting radioisotopes. The sensitivity, resolution and fast acquisition of images due to the solid-state detector technology makes this technology have a great impact in nuclear diagnosis [27,28]. For comparison, SPECT is widely available and costs less. The research for new radionuclides is on the way to attenuate the big limitation of PET/CT (uptake localization) [21]. The advancements in radiometal-based radiopharmaceuticals for PET can lead to improvements in radiometal designing research for SPECT radiometals (e.g., technetium) [27]. An important aspect is dose optimization of radiation, and this will individualize the use of either PET or SPECT (uses less radiation doses) technology and will have to be evaluated considering the patients risks, in order to provide quality scan pictures for relevant clinical decisions [29,30].

The search for molecular imaging agents is a current area of great interest for the unique imaging and molecular characterization of renal tumors, beyond the well-known used and approved histology of renal lesions [3,31]. The antibodies carrying radionuclides are targeting surface tumor antigens. Other small molecules and peptides, target specific cellular processes and can provide data on the subsequent histology of tumors within the kidney [32]. This molecular characterization of tumors can overcome the limitations of conventional imaging in order to identify the histological heterogeneity of renal tumors. Recently, the radiolabeled molecular antibody girentuximab and a mitochondrial imaging agent (^99m^Tc-MIBI) were heavily studied regarding the differentiation of renal lesions [33,34].

### 3.2. ^99m^Tc-Sestamibi SPECT/CT

Prior to the introduction of PET in clinical use, sestamibi radiopharmaceuticals were used to detect other types of tumors, such as breast cancer [35,36,37], the evaluation of mediastinal involvement in lung cancer patients [38], in multiple myeloma [39], the follow-up of brain glioma after chemotherapy [40] and ongoing and present use to characterize parathyroid adenomas [41,42].

In 2015, an initial trial performed by Rowe et al. identified that all oncocytomas demonstrated radiotracer uptake, but not the RCCs tumors [43]. One of the first studies that prospectively assessed the accuracy for the differentiation of oncocytomas and hybrid oncocytic/chRCC from other kidney lesions identified an overall sensitivity of 87.5% (95% confidence interval [CI], 47.4–99.7%) and a specificity of 95.2% (95% CI, 83.8–99.4%) [44]. In a pilot study, Tzortzakakis et al. aimed to characterize solid kidney lesions and the differentiation of oncocytomas from RCCs, and found that 91.6% of oncocytomas displayed positive uptake of ^99m^Tc-sestamibi [45]. Sheikhbahaei et al. assessed the addition of ^99m^Tc-sestamibi SPECT in order to increase the degree of diagnostic confidence to distinguish between benign and cancerous tissue, obtaining an area under the curve (AUC) of 0.60 for conventional imaging alone and 0.85 after reviewing ^99m^Tc-sestamibi (*p* for difference = 0.03) [46]. Due to the close uptake ratio cutoff for some of the tumors investigated so far, Jones et al. and Tzortzakakis et al. aimed to standardize uptake value both for the intra- and inter-observer agreement, to see if a quantitative SPECT/CT reconstruction method could be applied in the context of a clinical setting and if quantitative biomarkers could be validated for the evaluation of renal lesions. The improvement of separation between uptake ratios and a high intra-class correlation coefficient could discriminate further between different renal lesions [47,48]. Zhu et al. developed a study that aimed to assess if dual-phase ^99m^Tc-sestamibi SPECT/CT could identify benign or cancerous renal masses in a cohort of 147 patients and showed that benign tissues demonstrated a significantly elevated early relative uptake value and delayed relative uptake value than cancerous lesions (*p* < 0.0001) with high degree of sensitivity, specificity and accuracy (100% sensitivity, 94.8% specificity 94.8%, accuracy 95.3% and sensitivity 100%, specificity 96.3%, accuracy of 96.6%, respectively). In the context of dual phasing, both can differentiate between the renal lesions and the delay phase points a higher diagnostic accuracy [33]. A cost effective study has been also performed for this imaging tool and identified a positive cost effectiveness when compared with existing strategies [49]. Lately, Warren et al. started a pilot diagnostic accuracy study to evaluate ^99m^Tc-sestamibi SPECT/CT compared to a reference standard of histopathology and reported that the sensitivity and specificity to detect oncocytic/ch lesions from other RCCs was 100% (95% CI, 74–100%) and 100% (95% CI, 63–100%), respectively. Sensitivity and specificity to discriminate between benign versus malignant masses was 100% (95% CI, 54–100%) and 85.7% (95% CI, 57–98%), respectively [50]. Following the same rationale of detecting the accuracy of ^99m^Tc-sestamibi SPECT/CT in the diagnosis of renal lesions, compared with contrast enhanced CT only, Parihar et al. showed a sensitivity and specificity of SPECT/CT for the diagnosis of tumors of low malignant potential or benign to be 66.7%, and 89.5%, respectively, compared to 10% and 75% for contrast-enhanced CT, respectively [51]. Sistani et al. demonstrated that the ^99m^Tc-sestamibi SPECT/CT has the potential to characterize indeterminate renal lesions [52]. Viswambaram et al. determined the effectiveness of ^99m^Tc-sestamibi SPECT/CT to differentiate between malignant and benign renal lesions having a sensitivity of 89% (95% CI, 77–95%) and a specificity of 73% (95% CI, 45–91%). On the other hand, the authors found that the diagnostic accuracy was not improved when compared to visual interpretation alone [53]. Taken altogether, non-concerning lesions defined as oncocytic or benign tumors have very high sensitivity and specificity of being diagnosed with ^99m^Tc-sestamibi SPECT/CT when compared with only contrast-enhanced CT [51]. Asi et al. identified that all oncocytomas identified in their study had a high uptake of ^99m^Tc-sestamibi. There were 64 (71.11%) RCC lesions that had no uptake of the labeled radiotracer, resulting in a positive predictive value (PPV) and negative predictive value (NPV) of 60% and 91.3%, respectively, for identifying benign lesions [54]. Warren et al. reported results from a pilot diagnostic accuracy trial in patients with renal lesions aiming to assess, with the help of expert radiologists, the quantitative uptake of the radiotracer in the lesions compared to the healthy renal parenchyma and between different lesions subtypes, with excellent results on sensitivity, specificity in identifying ch lesions from RCCs and to differentiate benign from cancerous lesions [50]. Sistani et al. found in a prospective comparative study on 29 patients that the benign or oncocytic tumors compared to RCCs had a higher mean and maximum relative lesion uptake (*p* = 0.016 and 0.012, respectively) [52].

Presently, the recommendations from the EAU guidelines are that CT and MRI can accurately make a diagnosis of RCC, but cannot reliably distinguish oncocytoma and fat-free AML from malignant renal neoplasms [55]. However, the new imaging technique ^99m^Tc-sestamibi SPECT/CT show promising results to differentiate between benign and low grade RCC [56]. A summary of evidence is listed in Table 1.

The combination of presence or absence of sestamibi uptake and lesions’ density on SPECT/CT, the efforts to standardize the cut-off values, the very good results in terms of sensitivity, specificity and accuracy to distinguish between benign and malignant renal tissues could represent a novel imaging approach to stratify the risk of marginal/low risk [32] described kidney lesions on contrast-enhanced CT. The additional studies on health economics [49], additional validation [52,54] and the added value in renal lesions diagnosis (already approved for myocardial and parathyroid imaging [57]), ^99m^Tc-sestamibi SPECT/CT can make a fast transition into clinical practice for evaluating kidney lesions, with minimal costs.

### 3.3. Girentuximab PET/CT

The carbonic anhydrase IX (CAIX) antibody, found to have the ability to link on an antigen expressed on the surface of ccRCC [58], is present in over 95% of the ccRCC tumors [34]. It is not expressed in normal kidney parenchyma, cysts, papRCC or chRCC subtypes [59]. Girentuximab, a chimeric IgG1 monoclonal antibody [60], developed to target the CAIX antigen was studied with the purpose of diagnostic imaging along with PET [32,34]. Several radionuclides were used to label girentuximab for diagnostic therapeutic purposes [61]. Divgi et al. aimed to preoperatively assess, in an open-label pilot study, the iodine-124-labelled antibody chimeric G250 (^124^I-cG250) PET/CT) to accurately detect ccRCC with a 94% sensitivity, a NPV of 90%, and specificity and positive predictive accuracy of 100% [62]. Due to these results, the author designed a clinical study to assess the characterization of renal masses provided by ^124^I-cG250) PET/CT and found very good sensitivity and specificity (86.2% and 85.9%, respectively) and very good inter-reader and intra-reader agreement [63]. A 2018 multi-center trial that is an ongoing study was developed using ^89^zirconium-girentuximab (^89^Zr-girentuximab) instead of I^124-^girentuximab, because it may have high sensitive ability to detect ccRCC (ZIRCON trial, registered on ClinicalTrials.gov: NCT03849118) [64,65,66].

## 4. Discussion

The need for more comprehensive, accurate preoperative diagnosis of renal lesions that have the potential to progress and metastasize, in the setting of almost one fifth localized lesions being identified as benign in nature [9,10], is of great importance to minimize morbidity and renal function loss after the surgery of benign tumors [67]. The current research holds a positive premise to the identification of radiotracers and antibodies linked to different antigens of cancer cells [68], which can better distinguish between benign lesions, cancerous lesions and also different histological subtypes of RCC [3]. Being a novel area of investigation, most of the works are presented as pilot studies, with a limited number of enrolled subjects. Although this limitation can be applied, the results of trials investigating ^99m^Tc-sestamibi SPECT have very good accuracy in detection of benign tumors (83.3% up to 100% for oncocytoma) [33,44,54], showing that the technique has great diagnostic potential for these types of tumors. For AMLs, the accuracy was 100% [33,48], with one case of AML that had false-positive uptake [46] and 100% for HOCTs [44,47,48]. These AMLs and oncocytomas are rich in mitochondria [69], and therefore the uptake of the lipophilic ^99m^Tc-sestamibi is high in oncocytomas and AMLs, compared to malignant tissues where the multidrug resistance pump (performs wash-out of ^99m^Tc-sestamibi) is well expressed, making it good to discriminate oncocytomas from cancerous tissues [54,70].

For the chRCC subtype, we have identified a poorer performance in some studies (of just 50%) [44], but with excellent results in other research performed (100%) [50]. This high variance of results is probably due to the lack of standardization of protocols, time of exposure from injections that varies greatly between studies (from 1 to 7 days), and between molecules used (5 days for girentuximab or approximately 75 min for ^99m^Tc-sestamibi) [43,63].

RCC and its subtypes do not uptake the radiotracer, with the exception of chRCC. Two studies reported 50% false uptake [44,46], and one study showed slightly positive uptake of the radiotracer [45]. What is more important is that the sensitivity and specificity is very high in detecting benign compared to malignant lesions (ranging from 87.5% to 100% for sensitivity and 73% to 94.8% for specificity) [33,44,53]. Almost all studies have shown the ability of ^99m^Tc-sestamibi SPECT/CT to distinguish the benign lesions from malignant lesions, and this technique could be close to clinical implementation for diagnostic purposes in RCC, but there are some drawbacks to a generalized use. Girentuximab has a long stay in the main circulation of the body, therefore it is accumulated in the whole renal parenchyma [71], limiting the acquired quality of the images. The ^99m^Tc-sestamibi SPECT/CT can be performed relatively quickly after injection [43], limiting the long period and logistics to scan the patients. PET/CT has a greater image quality and time for slice acquisition [72], and used with a new modified ligand (^64^Cu XYIMSR-06) can shorten acquisition time compared to SPECT/CT [73]. Taken altogether, the techniques of SPECT/CT and PET/CT combined with molecular antibodies and radiotracers give new insights for the possibility of preoperatively distinguishing between benign and malignant lesions, and in some proportion to differentiate between subtypes of RCC. ^99^mTc-sestamibi is approved by the Food and Drug Administration (FDA) and European Medicines Agency (EMA) [74] to be used as a myocardial perfusion imaging molecule for use in SPECT for the identification of the risk of myocardial events [75], for assessment of breast, lung, thyroid, and head and neck benign or malignant tumors [76,77]. The girentuximab PET/CT holds great promise, but in this moment its use is limited to clinical trials [78]. A head-to-head comparison is not eligible due to the lack of data from clinical trials. Lately, there are new techniques evolving, such as the new long axial field-of-view PET/CT scanners, that are trying to overcome the disadvantage of the limited axial field of view of current scanners, and to provide an accurate dynamic scan of whole body tumors at the same time with short acquisition time [79,80,81].

There are no prospective, large, multi-center studies to validate the results obtained so far. There are no standardized protocols for image acquisition after the injection of radiotracers, no head-to-head comparison between SPECT/CT and PET/CT. These limit the application outside clinical trials. Validation and health economics studies will have to be performed before the worldwide clinical implementation of these techniques, but the future is here.

## 5. Conclusions

Nuclear medicine has helped clinicians in evaluating primary and secondary lesions and has lately returned with new and exciting insights with novel radiotracers to reinforce its diagnostic potential in renal carcinoma. The purpose is to limit the benign excised tumors and, subsequently, the morbidity and renal function loss in these patients. Further research is mandatory to validate the results and to clinically implement such diagnostic techniques in the context of precision medicine.

## Figures and Tables

**Figure 1 diagnostics-13-00593-f001:**
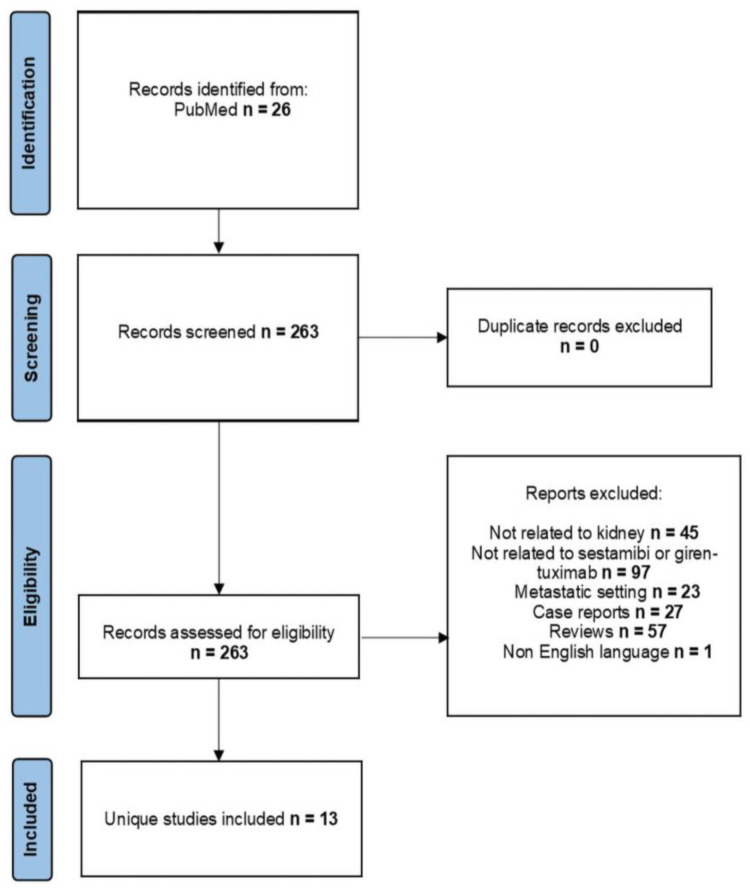
Flowchart of identified studies for analysis.

**Table 1 diagnostics-13-00593-t001:** Evaluation of studies assessing ^99m^Tc-sestamibi SPECT/CT in the diagnosis of renal cancer.

Authors	Study Type/No. Patients or Histological Diagnosis	Pathologic Diagnosis/Biopsy	Positive/Negative on ^99m^Tc-Sestamibi SPECT/CT, n (%)	Results
Rowe et al. [43]	Evaluation prospective study/6	Oncocytoma RCC	Yes, 3 (100%) No, 3 (100%)	Maximum radiotracer uptake similar/above adjacent renal parenchyma (average tumor uptake, 1.19; range, 0.85–1.78) Markedly photopenic relative to renal parenchyma average tumor uptake of 0.26 (range, 0.21–0.31)
Gorin et al. [44]	Prospective/50	Oncocytoma HOCTs chRCC RCC	Yes, 5 (83.3%) Yes, 2 (100%) Yes, falsely 2 (50%) No, 40 (100%)	^99m^Tc-sestamibi SPECT/CT correctly identified 5 of 6 (83.3%) oncocytomas and 2 of 2 (100%) HOCTs Good sensitivity of 87.5% (95% CI, 47.4–99.7%) and specificity of 95.2% (95% CI, 83.8–99.4%).
Tzortzakakis et al. [45]	Prospective/27	Oncocytoma HOCTs AML ccRCC papRCC chRCC ch/papRCC	Yes, 12 (91.6%) Yes, 3 (100%) Yes, 1 (100%) No, 0 No, 1 slightly positive (33.3%) No, 0 No, 0	91.6% of oncocytomas displayed positive uptake of ^99m^Tc-sestamibi
Sheikhbahaei et al. [46]	Prospective/48	Oncocytoma HOCTs AML RCC papRCC ccRCC Unclassified RCC chRCC	Yes, 6; (12.5%) False-positive 1 Yes, 2 (4.2%) Yes 2, (4.2%) False-positive results 1 No, 25 (52%) No, 4 (8.3%) No, 2 (4.2%) No, 1 (2.1) No, 4; (8.3%) Yes 2 falsely high uptake	AUC 0.60 to distinguish between benign and cancerous tissue for conventional imaging alone AUC 0.85 after reviewing ^99m^Tc-sestamibi (*p* for difference = 0.03)
Zhu et al. [33]	Prospective/147 patients and 148 histopathology results	Oncocytoma AML Malignant tumors	Yes, 4 (2.7%) Yes, 8 (5.4%) No, 124 (83.8%) Partial early uptake, 7 (5.1% from malignant tumors) Partial delayed uptake, 5 (3.7% from malignant tumors)	Higher ERUV and DRUV than malignant renal tumors (*n* = 136; both *p* < 0.0001). ERUV cutoff value of 0.53 helped to differentiate benign from malignant renal tumors (DRUV of 0.50), with sensitivity of 100%, specificity of 94.8%, and accuracy of 95.3% (DRUV sensitivity 100%, specificity 96.3%, accuracy of 96.6%) for the diagnosis of benign renal tumors.
Parihar et al. [51]	Retrospective/36 lesions	non-concerning lesions (oncocytic or benign tumors) ccRCC	Yes, 15 (41.7%) No, 13 (36.%)	The sensitivity and specificity of SPECT/CT for discriminating oncocytic or benign tumors = 66.7%, and 89.5%, respectively, compared to 10%, and 75% for CECT, respectively.
Asi et al. [54]	Prospective/90 lesions	Oncocytoma AML Chronic sclerosis Fibroma Hydatic cyst chRCC ch/papRCC All RCC	Yes, 10 (11.1%) No, 4 (4.5%) No, 2 (2.2%) No, 1 (1.1%) No, 1 (1.1%) Yes, 5 (5.6%) Yes, 3 (3.3%) No, 64 (71.1%)	PPV and NPV of 60% and 91.3%, respectively for identifying benign lesions.
Warren et al. [50]	Prospective, pilot diagnostic accuracy/19 lesions	Oncocytoma Oncocytic RCC chRCC ccRCC papRCC histology unknown	Yes, 6 (31.657%) Yes, 1 (5.3%) Yes, 1 (5.3%) No, 9 (47.4%) No, 2 (10.5%) No, 1 (5.2%)	Qualitative assessment for sestamibi-renal images categorized as avid or photopenic has been identified as 100%. Sensitivity and specificity of sestamibi-renal to detect ch lesions from other RCCs was 100% (95% CI 74 100%) and 100% (95% CI 63–100%), respectively. Sensitivity and specificity for discriminating benign versus cancerous lesions was 100% (95% CI 54–100%) and 85.7% (95% CI 57–98%), respectively.
Sistani et al. [52]	Prospective, comparative study/29 patients	Oncocytoma HOCT chRCC ccRCC papRCC cc/papRCC chRCC	Yes, 7 (24.1%) Yes, 1 (3.5%) Yes, 1 low uptake (3.5%) No, 15 (51.7%) No, 4 (13.8%) No, 2 (6.9%) No, 1 (3.4%)	The benign or oncocytic tumors compared to RCCs had higher mean and maximum relative lesion uptake (*p* = 0.016 and 0.012, respectively).
Viswambaram et al. [53]	Prospective/74 patients	Accuracy to detect malignant or benign lesions	49 (66.2%) 11 (14.8%)	Sensitivity of 89% (95% CI 77–95%) and a specificity of 73% (95% CI 45–91%) for detecting malignant or benign tumors.

AML = angiomyolipoma; AUC = area under the curve; ccRCC = clear cell renal cell carcinoma; CECT = contrast-enhanced computed tomography; chRCC = chromophobe renal cell carcinoma; CI = confidence interval; DRUV = delayed relative uptake value; ERUV = early relative uptake value; HOCT = renal hybrid oncocytic/chromophobe tumors; NPV = negative predictive value; pap = papillary; PPV = positive predictive value; PET-CT = positron emission tomography computed tomography; RCC = renal cell carcinoma; SPECT/CT = single photon emission computed tomography/computed.

## Data Availability

Not applicable.

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
