# Peer review of "Molecular Imaging Diagnosis of Renal Cancer Using 99mTc-Sestamibi SPECT/CT and Girentuximab PET-CT-Current Evidence and Future Development of Novel Techniques"

_diagnostics, 2023, doi:10.3390/diagnostics13040593_

Round 1
Reviewer 1 Report
The paper under evaluation is a systematic review focused on the role of SPECT with sestamibi vs. immunoPET with girentuximab PET-CT for the per-operative characterization of renal lesions.
Some considerations:
- the paragraph of Molecular Imaging is very poor. Please briefly explain the differences among SPECT (and SPECT/CT) and PET/CT. I would also suggest to mention that the authors will deal with a particular PET approach namely immunoPET.
- when the authors speak about Sestamibi, please briefly explain that this radiopharmaceutical has been widely used in oncology, for the imaging of different tumors such as breast cancer and others, before PET/CT’s implementation, and cite some relevant references (Anticancer Res. 2007 Jan-Feb;27(1B):557-62; doi: 10.1097/MNM.0000000000001259)
- In the section concerning girentuximab PET-CT the authors also mention a clinical reseach with 111In SPECT; I suggest to remove this reference since it has been carried out with a technology different from PET/CT.
- In the Discussion, it should be addressed that while sestamibi is an FDA and EMA approved radiopharmaceutical, girentuximab PET-CT is still experimental, therefore comparison among these 2 approaches is biased by the shortage of data concerning this specific PET imaging.
- Finally, I warmly suggest to more clearly underline that the implementation of some novel technologies, such as the new long axial field-of-view PET/CT scanners, might represent a game-changer in the field of immunoPET enabling dynamic studies and short-time acquisition protocols (cite doi: 10.1007/s00259-022-05777-x and doi: 10.1080/17434440.2022.2141111).
Author Response
The paper under evaluation is a systematic review focused on the role of SPECT with sestamibi vs. immunoPET with girentuximab PET-CT for the per-operative characterization of renal lesions.
We thank the reviewer for the valuable comments, insights and appreciation on the worked we had performed.
Some considerations:
- the paragraph of Molecular Imaging is very poor. Please briefly explain the differences among SPECT (and SPECT/CT) and PET/CT. I would also suggest to mention that the authors will deal with a particular PET approach namely immunoPET.
We thank the reviewer for these comments and we have adapted the text accordingly.
- when the authors speak about Sestamibi, please briefly explain that this radiopharmaceutical has been widely used in oncology, for the imaging of different tumors such as breast cancer and others, before PET/CT’s implementation, and cite some relevant references (Anticancer Res. 2007 Jan-Feb;27(1B):557-62; doi: 10.1097/MNM.0000000000001259)
We thank the reviewer for these comments and we have adapted the text and referenced it, accordingly as suggested.
- In the section concerning girentuximab PET-CT the authors also mention a clinical reseach with 111In SPECT; I suggest to remove this reference since it has been carried out with a technology different from PET/CT.
We thank the reviewer for this valuable suggestion and we have removed the text and the reference.
- In the Discussion, it should be addressed that while sestamibi is an FDA and EMA approved radiopharmaceutical, girentuximab PET-CT is still experimental, therefore comparison among these 2 approaches is biased by the shortage of data concerning this specific PET imaging.
We thank the reviewer for the comment and we had briefly included the suggested statement.
- Finally, I warmly suggest to more clearly underline that the implementation of some novel technologies, such as the new long axial field-of-view PET/CT scanners, might represent a game-changer in the field of immunoPET enabling dynamic studies and short-time acquisition protocols (cite doi: 10.1007/s00259-022-05777-x and doi: 10.1080/17434440.2022.2141111).
We thank the reviewer for the comment and suggestion and we had briefly included data on the topic mentioned.
Reviewer 2 Report
The article "Molecular Imaging Diagnosis of Renal Cancer using 99mTc-sestamibi SPECT/CT and girentuximab PET-CT-current evidence and future development of novel techniques" by Tataru et al. has a moderate level of novelty in terms of topics. However, the authors present well the current state-of-the-art of the subjects mentioned. The search of articles is reliable and the number may be considered sufficient for a brief review article. They also provide clear conclusions and critical analysis of the work done in the molecular imaging diagnosis of renal cancer using the specified techniques. In general, the article is well-written, but some minor corrections must be done in order for it to be accepted for publication.
"In United States it is estimated that kidney and renal pelvis cancers in 2019 will be 55 around 73,820 new cases and 14,770 new cancer related deaths [1].: in this sentence are mentioned the numbers of cancer that will appear in 2019, but 2019 as already passed. Please clarify.
In section 2 the authors specify that the works are from the last five years and after they say its for the last 10 years. Please clarify.
Line 131: clarify what AUC means.
Line 161: 0.89 and 0.73 should be in the form of percentages like earlier in the manuscript. There are other situations like this throughout the manuscript. It is suggested to provide consistency in the presentation of results from the works.
In the table, there are some values with two decimal numbers and others with only one. For the sake of consistency provide all values with either one or two decimal numbers.
Lines 214-217: the sentence is not clear, it must be re-written in order to be more fluid and clear.
Line 257: "the" is not necessary in this case.
Author Response
The article "Molecular Imaging Diagnosis of Renal Cancer using 99mTc-sestamibi SPECT/CT and girentuximab PET-CT-current evidence and future development of novel techniques" by Tataru et al. has a moderate level of novelty in terms of topics. However, the authors present well the current state-of-the-art of the subjects mentioned. The search of articles is reliable and the number may be considered sufficient for a brief review article. They also provide clear conclusions and critical analysis of the work done in the molecular imaging diagnosis of renal cancer using the specified techniques. In general, the article is well-written, but some minor corrections must be done in order for it to be accepted for publication.
We thank the reviewer for the valuable comments, insights and appreciation on the worked we had performed.
"In United States it is estimated that kidney and renal pelvis cancers in 2019 will be 55 around 73,820 new cases and 14,770 new cancer related deaths [1].: in this sentence are mentioned the numbers of cancer that will appear in 2019, but 2019 as already passed. Please clarify.
We thank the reviewer for this comment and we had adapted the text to the new updated 2023 cancer statistics and the reference, accordingly.
In section 2 the authors specify that the works are from the last five years and after they say its for the last 10 years. Please clarify.
We thank the reviewer for this comment and we had adapted the text accordingly (ten years).
Line 131: clarify what AUC means.
We thank the reviewer for this comment and we had explained the acronym as suggested.
Line 161: 0.89 and 0.73 should be in the form of percentages like earlier in the manuscript. There are other situations like this throughout the manuscript. It is suggested to provide consistency in the presentation of results from the works.
We thank the reviewer for this comment and we had changed to percentage accordingly and reviewed the whole text for consistency.
In the table, there are some values with two decimal numbers and others with only one. For the sake of consistency provide all values with either one or two decimal numbers.
We thank the reviewer for this comment and we had changed the percentage with one decimal, accordingly.
Lines 214-217: the sentence is not clear, it must be re-written in order to be more fluid and clear.
We thank the reviewer for this comment and we had removed the sentence due to inconsistency with the technology described in this section.
Line 257: "the" is not necessary in this case. We thank the reviewer for this comment and we had removed “the”.
Round 2
Reviewer 1 Report
The authors have properly addressed Reviewers' concerns.